# Outcome of the Use of Clinical Reasoning Alone in Dogs with Acute Thoracolumbar Myelopathy and/or Pain

**DOI:** 10.3390/ani14071017

**Published:** 2024-03-27

**Authors:** Bruno Scalia, Paul Freeman

**Affiliations:** Queen’s Veterinary School Hospital, University of Cambridge, Madingley Road, Cambridge CB3 0ES, UK; pf266@cam.ac.uk

**Keywords:** spinal cord injury, T3–L3, disc extrusion, disc herniation, IVDE, ANNPE, FCE, imaging, canine

## Abstract

**Simple Summary:**

Sudden onset hind limb weakness and incoordination with or without back pain is common in dogs. As certain diagnoses tend to present with a characteristic clinical picture, this study aimed to demonstrate the feasibility of managing canine patients with conditions such as intervertebral disc disease (IVDD) and so-called fibrocartilaginous embolism (FCE) or spinal stroke with clinical reasoning alone (i.e., without the aid of relatively costly three-dimensional imaging). By searching the database of the Queen’s Veterinary School Hospital (Cambridge), 139 dogs with initially suspected IVDD or FCE in their back were retrieved. The cases were divided into two groups based on suspected diagnosis. A total of 81% of cases with suspected compressive IVDD (not confirmed with imaging) had a successful outcome. A further 16% had the diagnosis confirmed with imaging and subsequent successful outcome, whilst just 3% had a different diagnosis or were euthanised without imaging. A total of 94% of suspected FCE or non-compressive IVDD cases had a successful outcome, and just one was euthanised due to failure to recover. Successful outcomes can be obtained by using clinical reasoning alone in most dogs with suspected acute IVDD or FCE.

**Abstract:**

Acute thoracolumbar myelopathy is a common neurological presentation in dogs. Although certain spinal conditions present with characteristic clinical pictures, managing such cases with clinical reasoning alone (i.e., without cross-sectional imaging) has never been explored. The aim of this study was to define the outcome of patients with suspected intervertebral disc extrusion (IVDE), ischaemic myelopathy (IM) or acute non-compressive nucleus pulposus extrusion (ANNPE) that were managed with clinical reasoning alone. The database of the Queen’s Veterinary School Hospital (Cambridge) was searched for paraparetic or paraplegic dogs with suspected IVDE or IM/ANNPE that were initially managed medically without undergoing imaging. Clinical presentation and outcome were recorded. If cross-sectional imaging was subsequently performed, information about the final diagnosis was collected and compared with the initially suspected diagnosis. A total of 123 IVDE cases were collected: 81% had a successful outcome with no imaging performed; 16% had IVDE confirmed with imaging and successful outcome with surgery or medical management; and just 3% were found to have an alternative diagnosis or were euthanised without imaging. A total of 16 IM/ANNPE cases were collected: 94% had a successful outcome, and one dog was euthanised. Successful outcomes can be obtained by using clinical reasoning alone in most dogs with suspected IVDE or IM/ANNPE.

## 1. Introduction

Acute spinal cord injury (SCI) represents a very common neurological problem in dogs [1,2]. The source of damage to the spinal cord can arise from within or outside of the parenchyma. The latter is more common, especially in the thoracolumbar spinal cord region of dogs, with intervertebral disc extrusion (IVDE) representing the most frequent cause of acute SCI in this region for this species [3]. Incidence and mortality of thoracolumbar IVDE for commonly affected breeds, such as Dachshunds, are as high as 41% and 4.2%, respectively [2]. There is also a known gender and age predisposition, with males being at higher risk and a higher likelihood of IVDE occurrence between 4 and 6 years of age [2,4]. When presented with a canine patient showing clinical signs of myelopathy localising to the region between the third thoracic and third lumbar spinal cord segments (T3–L3), many neurologists use an acronym such as “VITAMIND” to generate a list of differential diagnoses on which further diagnostic steps can be based [5], following the principles of clinical reasoning. Clinical reasoning has been defined as “a skill, process, or outcome wherein clinicians observe, collect and interpret data to diagnose and treat patients. It entails both conscious and unconscious cognitive operations interacting with contextual factors such as the patient’s unique circumstances and preferences and the characteristics of the practice environment” [6]. Clinical reasoning has been shown to be relatively reliable in veterinary spinal conditions because specific aetiologies tend to present with typical clinical features, such as speed of onset and clinical progression [3,7]. When it comes to thoracolumbar IVDE, although these patients can present with a combination of clinical signs ranging from spinal pain to paraplegia [8], they are most commonly small, young/middle-aged dogs of chondrodystrophic breeds such as Dachshunds, with an acute and deteriorating history of painful myelopathy [3]. On the other hand, ischaemic myelopathy (IM) or acute non-compressive nucleus pulposus extrusion (ANNPE) typically present with an acute/peracute and improving history of non-painful, lateralising myelopathy [3,5], sometimes following strenuous exercise [9,10], and with some predilection for males [11,12] and for certain breeds such as miniature Schnauzer and Staffordshire bull terrier [13,14]. Similarly to IVDE, clinical signs occur at around 5–6 years of age, and prognosis is generally very good as long as nociception is preserved [11,12]. The process of clinical reasoning is commonly and perhaps sometimes subconsciously adopted by clinicians [15,16] and can assist with the successful management of such cases, especially where financial concerns may limit diagnostic work-up. The cost of veterinary care can be prohibitive for some owners [17,18], and a risk of potentially unnecessary euthanasia exists if veterinary surgeons are not aware of the likely differential diagnoses and prognoses [2]. It may also be the case that although cross-sectional imaging such as magnetic resonance (MRI) and computed tomography (CT) represents the gold standard for the diagnosis of acute SCI [19,20,21,22,23,24,25,26], the patient may not benefit from the mere confirmation of the suspected diagnosis if this does not affect the treatment plan [27,28,29].

At the authors’ institution, dogs presenting with a clinical picture strongly suggestive of IVDE or IM/ANNPE and for whom surgical treatment is not immediately anticipated are frequently managed without cross-sectional imaging. Careful monitoring for the progression of signs has been found to be a valuable ‘test’ that can guide the clinician, modify the differential diagnosis list, and sometimes change the initial plan [15]. To the authors’ knowledge, the outcome of patients managed following such a rationale has not previously been explored. The questions this study aims to answer are the following: in dogs presenting with acute T3–L3 myelopathy treated conservatively for presumed IVDE, IM or ANNPE, what proportion does not improve satisfactorily and subsequently requires three-dimensional imaging? For how many of these is the final diagnosis different from the initially suspected diagnosis? Finally, what is the outcome for all these patients?

## 2. Materials and Methods

### 2.1. Case Identification

Cases were identified retrospectively by searching the medical records of the Queen’s Veterinary School Hospital, University of Cambridge (QVSH) between 2017 and 2022 for the words “paraparetic”, “paraparesis”, “paraplegic”, and “paraplegia”. Dogs were included if they met the following criteria: T3–L3 spinal cord segments neuroanatomical localisation, peracute or acute onset of signs, most likely differential diagnosis of IVDE or IM/ANNPE, and no cross-sectional imaging (CT/MRI) carried out the same day of the initial consultation. The onset of signs was judged qualitatively based on owner reporting. Specifically, onset was considered peracute if a specific day and time of onset could be determined, while it was considered acute if the signs started on a specific day, but a precise time of the day could not be established. IM and ANNPE were considered as a single entity as they are clinically indistinguishable. Signalment, history, onset, progression, presence of pain, lateralisation of signs and neuroanatomical localisation were used to define the most likely differential diagnosis [3,5,30]. Dogs who had spinal radiographs before referral were not excluded. Follow-up was obtained by searching the medical records or contacting the referring veterinary surgeons or the owners. Cases were excluded if medical records were incomplete and follow-up of at least two weeks was not available. Dogs who were paraplegic with absent pelvic limb and tail deep pain perception at presentation were also excluded as they would not be normally managed with clinical reasoning alone and would undergo cross-sectional imaging and decompressive surgery soon after presentation [31].

### 2.2. Data Collection

Medical records were evaluated for age, sex, breed, owner complaint, time from onset of signs to presentation to the QVSH, onset of signs as reported by owner (acute/peracute), progression (deteriorating/static/improving), presence of pain at presentation (yes/no), lateralisation of neurological signs (yes/no), presence of pre-referral radiographs (yes/no), concurrent/previous medical issues, neurological grade [defined using the modified Frankel scale [32,33] (Table 1)], most likely differential diagnosis (IVDE or IM/ANNPE), hospitalisation after the consultation (yes/no) and follow-up time. Dogs presenting within less than 24 h from the onset of signs were classified as presenting 1 day from the onset. Neurological signs were deemed lateralised if postural reactions, spinal reflexes and/or gait were reported to be asymmetrically abnormal. For dogs who did not improve, the time between initial presentation and second consultation (or imaging, if not discharged from the hospital), neurological grade at that stage, final diagnosis, treatment received and whether improvement in voluntary movement and pain was documented at follow-up were also recorded.

### 2.3. Outcome Groups

The outcome was considered successful if voluntary pelvic limb movement improved (non-ambulatory dogs becoming ambulatory paraparetic or normal; ambulatory dogs becoming less paraparetic or normal) and pain improved within 2–8 weeks from the initial consultation. This was defined based on clinician assessment and owner reporting. The outcome was considered unsuccessful if dogs did not improve or deteriorated neurologically (worsened gait and/or pain). Cases were then divided into the following three groups:Group 1: successful outcome achieved without undergoing cross-sectional imaging and with medical treatment only.Group 2: initially unsuccessful outcome after receiving medical treatment leading to cross-sectional imaging being carried out. The initial presumed diagnosis matched the imaging diagnosis, and eventually, the dogs had improved voluntary pelvic limb motor function and pain and, hence, a successful outcome.Group 3: unsuccessful outcome after receiving medical treatment, leading to cross-sectional imaging being carried out. The initial presumed diagnosis did not match with the imaging diagnosis. Cases where a final diagnosis was not achieved but the outcome was unsuccessful (e.g., dogs who remained static or were euthanised without imaging) were also included in this group.

Medical treatment consisted of rest and analgesia (± intravenous fluid therapy). A combination of non-steroidal anti-inflammatory drugs PO, gabapentin 8–15 mg/kg PO TID and paracetamol 10–15 mg/kg IV or PO TID was used as analgesia. Methadone 0.1–0.2 mg/kg IV every four hours was administered initially if required and slowly reduced and discontinued when the initial signs of pain subsided.

## 3. Results

### 3.1. Study Population

Inclusion criteria were satisfied by 141 dogs (one dog suffered suspected IVDE twice in the study timeframe; as these events happened with an interval of two months with full recovery in between, these were considered as separate occurrences, and the dog was counted as two separate individuals). Two dogs were excluded because of a lack of follow-up. A total of 139 cases were included in the study. A total of 123 and 16 dogs were included in the presumed IVDE and the presumed IM/ANNPE groups, respectively. The mean age at presentation was 6.2 years (range: 1.8–14.5). Of the 139 included dogs, 57 (41%) were male neutered, 51 (37%) were female neutered, 20 (14%) were male entire and 11 (8%) were female entire. The breeds represented in this study are detailed in Table 2 for each outcome group. Owner complaints for both presumed IVDE and IM/ANNPE are summarised in Table 3.

### 3.2. Presumed IVDE

One hundred twenty-three dogs were treated for presumed IVDE. Of these patients, 100/123 (81%, with 95% confidence interval (CI) 73–87) belonged to group 1, 19 (16%—CI: 10–23) to group 2, and 4 (3%—CI: 1–8) to group 3 (Figure 1). Ultimate successful outcome (groups 1 and 2) was achieved in 119/123 patients (97%—CI: 92–98) (Figure 1). If deterioration was detected and an imaging diagnosis was achieved (group 2 and two dogs of group 3), the initially suspected diagnosis was correct in 90% (18/20) of cases.

#### 3.2.1. Group 1

The main findings are summarised in Table 4. A specific incident such as slipping, falling or jumping on/off furniture was reported in 72% (8/11) of dogs with peracute onset of signs. Previous episodes of suspected thoracolumbar IVDE managed medically were reported in 7 dogs; these had taken place between 2 and 12 months before the current presentation. Six cases had previously had spinal surgery (4 hemilaminectomies for IVDE, one spinal stabilisation to treat a spinal fracture, and one spinal stapling to treat a subarachnoid diverticulum). Three dogs developed suspected chronic pain and were kept on long-term analgesia.

#### 3.2.2. Group 2

The main pre-imaging findings are summarised in Table 4. Two dogs had previously undergone a hemilaminectomy to treat IVDE one and four years before; 2 dogs had previously been treated medically for suspected or confirmed IVDE one and three years before. IVDE was the suspected diagnosis in all cases. The mean time between initial consultation and diagnostic imaging was 6.84 days (range: 1–28). Neurological grade at the time of diagnostic imaging was specified in 16 cases and ranged between 1 and 5 (mean 2.75). All patients underwent an MRI scan and were diagnosed with IVDE between and including T10–T11 and L3–L4 intervertebral discs. Surgery was performed in 18/19 patients, while 1 patient was managed conservatively.

#### 3.2.3. Group 3

A final diagnosis was achieved in two dogs.

Dog 1: 11-year-old female neutered crossbreed with a six-day history of acute, deteriorating, painful and lateralised poorly ambulatory paraparesis. Medical treatment for suspected IVDE was established. However, the dog presented four weeks later with multifocal neurological signs and was diagnosed with meningoencephalomyelitis of unknown origin (MUO).Dog 2: 8-year-old male neutered Border Terrier with a 2–3 weeks history of acute, improving, non-painful and lateralised ambulatory paraparesis. Following four weeks of medical treatment for suspected IVDE, the dog became non-ambulatory paraparetic and was subsequently diagnosed with a right T10 malignant peripheral nerve sheath tumour (PNST).

A final diagnosis was not achieved in two dogs.

Dog 3: 6-year-old male neutered Pug with a one-day history of acute onset ambulatory paraparesis that improved following administration of 1.3 mg/kg prednisolone. Despite crate rest and 0.7 mg/kg prednisolone SID, the dog’s gait slowly deteriorated; he developed urinary and faecal incontinence and was eventually euthanised.Dog 4: 7-year-old male neutered Dachshund with a two-week history of acute, deteriorating, non-painful, non-lateralised ambulatory paraparesis. Despite strict rest, the dog deteriorated to a non-ambulatory status, requiring manual bladder expressions. The owners were not keen on spinal surgery, and he was subsequently euthanised 4–6 weeks later.

### 3.3. Presumed IM/ANNPE

Sixteen dogs were treated for presumed IM/ANNPE. Successful outcome (group 1) was achieved in 15 (93.8%—CI: 67–99) patients, whilst only 1 case (6.2%—CI: 0.3—32.3) had an unsuccessful outcome (group 3). No dog met the criteria for group 2 (Figure 2).

#### 3.3.1. Group 1

The main findings are summarised in Table 4. A specific incident, such as falling or yelping while running or after jumping, was reported in 10/15 (66%) cases. Previous/concurrent medical issues were reported in 3 dogs (previous bilateral total hip replacement, leishmaniasis and demodicosis).

#### 3.3.2. Group 3

Dog 5: 13-year-old male neutered Staffordshire bull terrier with a one-day history of acute, improving, painful and lateralised non-ambulatory paraparesis. No pre-referral spinal radiographs were available, and osteoarthritis was the only other known concurrent health concern. Hospitalisation was declined by the owner, and the dog was euthanised the day after presentation due to neurological deterioration.

## 4. Discussion

Our findings suggest that clinical reasoning alone can lead to a successful outcome in many ambulatory dogs whose clinical presentation is compatible with what is known to be typical for thoracolumbar IVDE or IM/ANNPE.

Different concerns were expressed by owners about IVDE and IM/ANNPE. As previously reported [34], the main clinical complaint in IVDE reflected a pain syndrome, while this was not appreciated for IM/ANNPE. Fibrocartilaginous embolism myelopathy (FCE) and ANNPE can be painful over the first few hours after onset, and it is possible that discomfort was not appreciated in these dogs because of the delay between onset of signs and presentation to the clinician [9,25,35]. Moreover, interestingly, an incident just before or at the time of onset was reported in 68.7% of IM/ANNPE cases, as opposed to 8.9% in IVDE. Although risk factors associated with IVDE have been described in Dachshunds [34,36], research on incidents around the onset of IVDE is currently lacking.

Fourteen of our patients underwent spinal radiographs at their local veterinary practice. Although including dogs with spinal radiographs means that clinical reasoning was aided by imaging to a certain degree, the diagnostic sensitivity of radiographs without myelography for IVDE is low [37,38,39]. Therefore, although spinal radiographs can be useful to rule out some spinal neoplasias [40] and potentially increase the index of suspicion for IVDE [39], clinical reasoning was still the most valuable tool utilised in the management of these cases [15]. Additionally, in the authors’ experience, radiographs are frequently performed before referral. Therefore, this scenario reflects real life in referral practice.

Recovery from SCI can span weeks [8], and our IVDE group 2 dogs underwent cross-sectional imaging 6.84 days after initial presentation. As this was followed by surgery in 18/19 cases, it is unknown whether these 18 patients would have fully recovered if medical management had been continued for longer. A prospective study allowing more time before surgical intervention may help answer this question.

A previous spinal condition was reported in 13% of group 1 IVDE cases. Dogs can suffer long-term neurological deficits after spinal cord injury [41,42,43,44]. Therefore, it is possible that the neurological examinations of these dogs were impacted by previous IVDE or spinal surgeries. These cases were not excluded from our study as the time between the previous SCI and presentation was at least two months (mean 7.7 months). However, re-exacerbation of spinal pain secondary to a problem other than IVDE (e.g., implant failure) could not be excluded for a small number of cases. For instance, one dog presented with a one-week history of acute thoracolumbar pain and worsening paraparesis, having undergone the removal of spinal stapling implants placed for articular facet dysplasia and subarachnoid diverticulum 14 months previously. The two instances were considered separate due to the considerable time between them, during which time the dog’s condition had remained stable and the acute onset of the suspected IVDE episode. Another dog presented with a similar history, 4 years after having L5 fracture stabilisation; in this case, again, the presenting signs, history, and neuroanatomical localisation led to implant failure being considered very unlikely.

In IVDE, grade 2 (ambulatory paraparesis) was the most common degree of neurological impairment in all groups. This proportion differs from previously published case series [45,46,47] and implies a selection bias towards less severely affected cases in our population. Clinicians and owners may have been more comfortable proceeding with conservative management based on a suspected and unconfirmed diagnosis for patients that were still ambulatory. It would be interesting to see the results of a similar study involving a higher proportion of more severely affected dogs or to compare the outcome of a population similar to ours, which, however, undergoes cross-sectional imaging before medical management.

Depending on several factors, such as time of arrival at the hospital and owners’ preference, some of our patients were initially hospitalised and reassessed the following day. If an improvement was noticed, regardless of the initial grade at presentation, conservative management was pursued. If patients remained static, at least two weeks (or less, in case of deterioration) of medical management were generally performed before considering the outcome to be unsuccessful for dogs with neurological grade 1 or 2 (ambulatory paraparetic). The timescale for reviewing decision-making was generally 5–7 days (less in case of deterioration) for dogs with neurological grade 3 (non-ambulatory paraparetic). If the patient deteriorated, imaging was normally performed immediately (group 2). This approach stems from the concept that SCI from IVDE derives from a combination of different degrees of compression and contusion [8], and if the latter is the most significant factor in the patient’s clinical signs, then these will gradually improve as the contusion subsides over time. If the patient does not improve or remains painful, then the ‘compressive’ component of the IVDE may be more significant in that patient, and decompressive surgery may be more appropriate. This is also a relatively safe approach as the timing between the onset of signs and surgery does not seem to affect the outcome [48,49]. The only grade 4 dog with suspected IVDE in our study may have been initially managed medically due to time of arrival to the hospital or financial constraints and eventually underwent imaging and surgery 5 days after presentation due to lack of improvement.

Almost all our non-ambulatory patients were hospitalised and reassessed the following day (16 from both groups 1 and 5 from group 2 out of 23 non-ambulatory patients). This approach potentially allows the clinician to use progression of signs as a test in order to refine the list of differential diagnoses and the plan. This might be particularly helpful since, if these dogs were kept at home, owners may not appreciate subtle changes in their dog’s neurological status, which a neurology specialist or resident in training hopefully would see, which might significantly affect the plan. Hospitalisation rather than discharge to monitor at home was pursued more frequently with worse neurological grades both in presumed IVDE and IM/ANNPE. This reflects the concern that particularly non-ambulatory dogs affected with IVDE might deteriorate [50] and require surgical intervention. The authors feel it is often appropriate to use this approach in dogs that are non-ambulatory as well as less severely affected dogs, albeit only after a full discussion of the potential risks of deterioration with the animals’ owners and an explanation that the latest ACVIM Consensus suggests a surgical approach may be more suitable for non-ambulatory dogs [31].

Three dogs in IVDE group 1 developed suspected chronic back pain. Due to the retrospective nature of this study, this number may not reflect the real prevalence of this problem in our population. Chronic neuropathic back pain can occur in 15% of dogs after hemilaminectomy for IVDE [41]. However, further studies are needed to investigate this phenomenon in dogs that are managed medically.

The main limitations of this study are its retrospective nature and relatively small populations, especially in groups 2 and 3. The interpretation of written reports may have led to misunderstandings of the clinician’s ideas and intentions. The keywords used to research our medical records inherently excluded almost all dogs who presented with spinal pain in the absence of motor deficits. Equally, limited conclusions can be drawn on more severely affected dogs due to the overall paucity of non-ambulatory dogs. Lastly, although the mean follow-up time was 22 weeks for IVDE group 1 and 15.8 weeks for IM/ANNPE group 1, the minimum follow-up time was two weeks; unexpected variations of some dogs’ neurological status after two weeks (making other differential diagnoses more likely) cannot be excluded.

## 5. Conclusions

Ambulatory dogs with a clinical picture which generates IVDE as the most likely differential diagnosis have around an 80% chance of having a successful outcome when treated medically for suspected IVDE without the aid of cross-sectional imaging. If deterioration is seen following initiation of medical management but IVDE is subsequently confirmed, a successful outcome may be obtained in all cases. In this study, only 3% of suspected IVDE patients eventually received a different diagnosis or were euthanised without undergoing cross-sectional imaging. When patients underwent MRI due to deterioration, the final diagnoses were different from the initially suspected IVDE diagnosis in only 2/21 cases. At the same time, dogs with a clinical picture compatible with IM/ANNPE have a 94% chance of having a successful outcome even without cross-sectional imaging. The authors recommend the use of clinical reasoning with the aid of acronyms such as VITAMIND while keeping in mind the patient in their entirety to avoid potentially unnecessary tests and procedures, which may prove unaffordable for some owners and may not affect the plan for some patients.

## Figures and Tables

**Figure 1 animals-14-01017-f001:**
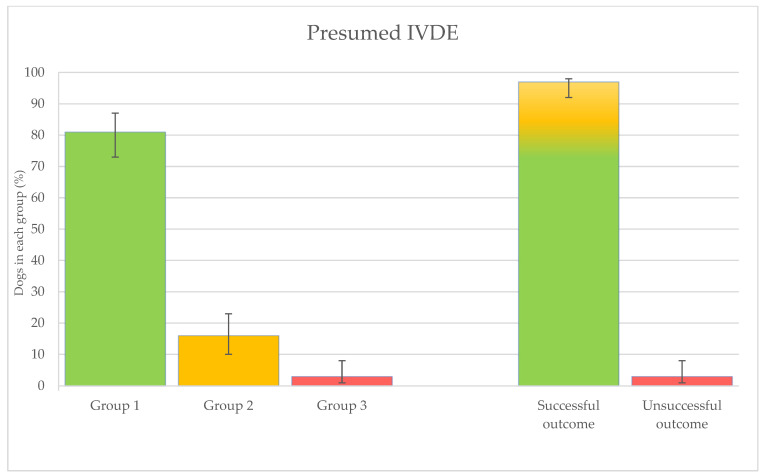
Percentages of dogs treated for presumed IVDE for each group and outcome.

**Figure 2 animals-14-01017-f002:**
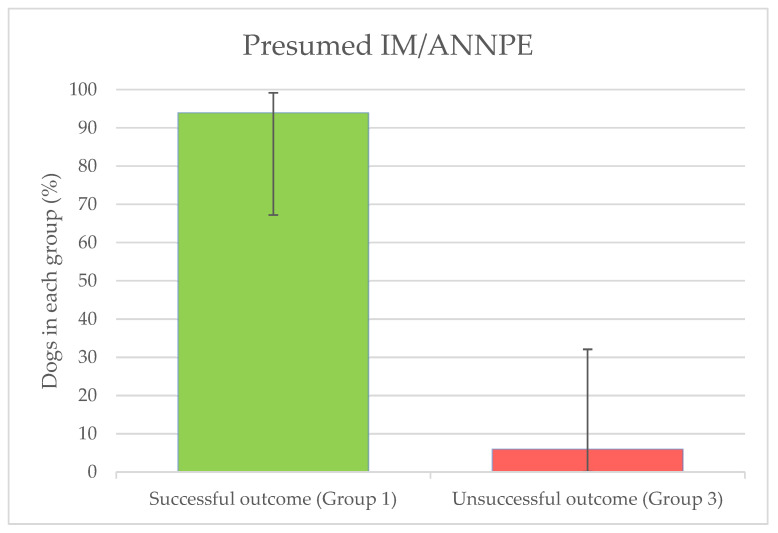
Percentages of dogs treated for presumed IM/ANNPE who had a successful (group 1) and unsuccessful (group 3) outcome.

**Table 1 animals-14-01017-t001:** Association between neurological grade and neurological status.

Neurological Grade	Neurological Status
1	Spinal pain only—normal gait
2	Ambulatory paraparesis
3	Non-ambulatory paraparesis
4	Paraplegia with intact pelvic limb and tail deep pain perception
5	Paraplegia with absent pelvic limb and tail deep pain perception

**Table 2 animals-14-01017-t002:** Number of dogs included in each outcome group for each breed.

	Presumed Intervertebral Disc Extrusion (IVDE)			Presumed Ischaemic Myelopathy/Acute Non-Compressive Nucleus Pulposus Extrusion (IM/ANNPE)	
	Group 1	Group 2	Group 3	Group 1	Group 3
Dachshund	41	10	1		
French Bulldog	18	2			
Crossbreed	11	1	1	1	
Staffordshire Bull Terrier	3			5	1
Cockapoo	4	2			
Pug	3		1		
Cocker Spaniel	4				
Border Collie				2	
Border Terrier	1		1		
Jack Russell Terrier	2				
Labrador	1			1	
Maltese		2			
American Bulldog				1	
American Cocker Spaniel		1			
Basset Hound	1				
Beagle	1				
Chihuahua	1				
Corgi	1				
Dalmatian	1				
Dandie Dinmont		1			
English Bulldog	1				
German Shepherd				1	
Great Dane				1	
Irish Water Spaniel				1	
Miniature Schnauzer	1				
Nova Scotian Retriever	1				
Schnauzer				1	
Shih Tzu	1				
Springer Spaniel	1				
Welsh Springer Spaniel	1				
Whippet				1	
Yorkshire Terrier	1				

**Table 3 animals-14-01017-t003:** Concerns expressed by owners at the time of the initial consultation.

Owner Complaint ^1^	Presumed IVDE	Presumed IM/ANNPE
Abnormal gait (no reported inciting event)	82.1% (101/123)	31.3% (5/16)
Pain (no specific information available)	33.3% (41/123)	
Lethargy	16.2% (20/123)	
Hunched posture	9.7% (12/123)	
Abnormal gait after jumping/running/falling	8.9% (11/123)	68.7% (11/16)
Pain when picked up	8.1% (10/123)	
Reluctance to move/jump	6.5% (8/123)	
Vomiting, diarrhoea, inappetence	4.8% (6/123)	
Incontinence	3.2% (4/123)	
Shivering	3.2% (4/123)	
Restlessness	2.4% (3/123)	
Spontaneous vocalisation	2.4% (3/123)	
Constipation	0.8% (1/123)	
Inability to lift tail when defecating	0.8% (1/123)	
Panting	0.8% (1/123)	

^1^ Some owners mentioned multiple concerns for the same patient.

**Table 4 animals-14-01017-t004:** Main findings and comparisons between IVDE group 1, IVDE group 2, and IM/ANNPE group 1.

		IVDE Group 1 (100 Dogs)	IVDE Group 2 (19 Dogs)	IM/ANNPE Group 1 (15 Dogs)
Mean age at presentation (years)		6.3 (range 1.8–14.5)	5.9 (range 2.16–13)	5.5 (range 1.33–10.4)
Mean time to presentation (days)		9.2 (range 1–90)	6.9 (range 1–28) *	2 (1–10)
Onset	Acute	87%	100% (19/19)	6.6% (1/15)
	Peracute	12%	0% (0/19)	93% (14/15)
	*Not stated*	1%	0% (0/19)	0% (0/15)
Progression of signs	Deteriorating	27%	74% (14/19)	6.6% (1/15)
	Static	13%	5% (1/19)	6.6% (1/15)
	Improving	53%	0% (0/19)	80% (12/15)
	*Not stated*	7%	21% (4/19) **	6.6% (1/15)
Spinal pain	Yes	74%	84% (16/19)	6.6% (1/15)
	No	23%	11% (2/19)	93% (14/15)
	*Not stated/equivocal*	3%	5% (1/19)	0% (0/15)
Lateralisation (yes)		64%	53% (10/19)	86% (13/15)
Spinal radiographs (yes)		9%	5% (1/19)	27% (4/15)
Neurological grade	1	1%	0% (0/19)	0% (0/15)
	2	86%	74% (14/19)	73.3% (11/15)
	3	13%	21% (4/19)	13.3% (2/15)
	4	0%	5% (1/19)	13.3% (2/15)
	5	0%	0% (0/19)	0% (0/15)
Mean neurological grade		2.1	2.3	2.4
Hospitalisation (yes)		47%	74% (14/19)	27% (4/15)
Mean neurological grade	Hospitalised	2.3	2.4	3.0
	Not hospitalised	1.9	2.0	2.2
Mean follow-up in weeks		22.0 (range: 2–260)	19.8 (range: 4–150)	15.8 (range: 2–96)

* Not specified in one dog. ** 3 of these dogs had onset of signs within 24 h from presentation.

## Data Availability

The raw data supporting the conclusions of this article will be made available by the authors on request.

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
