# Peer review of "Outcome of the Use of Clinical Reasoning Alone in Dogs with Acute Thoracolumbar Myelopathy and/or Pain"

_animals, 2024, doi:10.3390/ani14071017_

Round 1

Reviewer 1 Report

Comments and Suggestions for Authors

Thank you, authors, for your insightful article on acute spinal cord injury (SCI), a prevalent neurological problem. The primary focus was to delineate the outcomes of patients presenting with suspected IVDE, IM, or ANNPE, managed solely through clinical reasoning. The proposed approach, grounded in patient signalment, semeiotics, and differential reasoning using the acronym vitamin D, undoubtedly forms the foundation of a neurologist's rationale. However, it is crucial to recognise that imaging is essential for lesion characterisation.

In my understanding, CT or MRI scans were used for patients who showed no improvement after medical treatment, remained stable or worsened. I am perplexed by the assertion that outcomes are assessed solely through clinical reasoning, as a thorough clinical evaluation, complemented by imaging, appears pivotal for a precise diagnosis. The absence of statistical analysis in the data presentation is noteworthy, as merely reporting percentages lacks a more structured examination. I recommend the inclusion of statistical analysis to enhance the study's robustness.

The discussion (lines 257-259) and conclusion (334-336) mention that cross-sectional imaging was unnecessary in 80% of cases. To provide clarity, it is essential to correlate this statistic with the neurological status of patients at the time of referral.

The tables and references are commendable and up-to-date. However, specific points warrant attention:

- Line 116: The article lacks information on the evaluator's experience level. It is crucial to specify whether the same person performed the re-evaluation and elaborate on the experience levels of those involved.

- Lines 128-129: Explicitly stating the number of days is necessary, and in case of deterioration, the possibility of a patient transitioning to a different category should be addressed. For instance, if a grade 2 patient deteriorates to grade 3, the article should clarify the duration of conservative therapy before revisiting decision-making. Additionally, the authors should clarify whether an MRI or CT scan was conducted in such cases.

- Line 132: The rationale behind waiting four or five days before performing an MRI or CT scan for a grade 4 patient needs clarification. The same query applies to grade 5 patients. Providing insights into the decision-making process and timing of scans is crucial for a comprehensive understanding.

In conclusion, while the article offers valuable insights, addressing the above concerns would contribute to a more comprehensive and transparent presentation of the study's findings.

Author Response

Thank you, authors, for your insightful article on acute spinal cord injury (SCI), a prevalent neurological problem. The primary focus was to delineate the outcomes of patients presenting with suspected IVDE, IM, or ANNPE, managed solely through clinical reasoning. The proposed approach, grounded in patient signalment, semeiotics, and differential reasoning using the acronym vitamin D, undoubtedly forms the foundation of a neurologist's rationale. However, it is crucial to recognise that imaging is essential for lesion characterisation. In my understanding, CT or MRI scans were used for patients who showed no improvement after medical treatment, remained stable or worsened. I am perplexed by the assertion that outcomes are assessed solely through clinical reasoning, as a thorough clinical evaluation, complemented by imaging, appears pivotal for a precise diagnosis.

  • Thank you very much for taking the time to review our work and provide insightful comments and suggestions to improve its quality and clarity. The purpose of this study was to evaluate the efficacy of using clinical reasoning alone without initial imaging, which is a very common scenario in general practice, and whilst of course we recognise the value of imaging and the fact that this is the only way to confirm the diagnosis (as described in the introduction lines 78-80), our aim was to report pragmatic outcomes in terms of animal recovery and confirmation of a precise diagnosis was necessarily precluded by the study design. We feel this is a valid investigation since it represents and reports a process which occurs on a daily basis, and we felt would benefit from some analysis.

The absence of statistical analysis in the data presentation is noteworthy, as merely reporting percentages lacks a more structured examination. I recommend the inclusion of statistical analysis to enhance the study's robustness.The discussion (lines 257-259) and conclusion (334-336) mention that cross-sectional imaging was unnecessary in 80% of cases. To provide clarity, it is essential to correlate this statistic with the neurological status of patients at the time of referral.

  • Whilst we completely understand the reviewer’s concerns, following discussion with our statistician we feel strongly that further statistical analysis of the results would not add to the robustness or clarity of the study. We present descriptive statistics which we feel are appropriate to the main question the study is posing, and feel that an attempt to create significance and p-values around the results would not be helpful given our study design and aims. We understand the reviewer’s concern over correlation of neurological status with diagnosis (presumed or confirmed), but the aim of the study was to report a pragmatic outcome not related to initial neurological presentation, so we would request the reviewer to consider accepting the results reporting as presented given the other revisions.

- Line 116: The article lacks information on the evaluator's experience level. It is crucial to specify whether the same person performed the re-evaluation and elaborate on the experience levels of those involved.

  • The re-evaluations were performed by residents working under the supervision of specialists or specialists themselves. They were not always performed by the same person, hence the importance of owners’ perspectives in some cases to understand whether the patient improved or not, especially for patients that remained within the same neurological grade at last follow-up (e.g. a patient who improves from markedly to mildly paraparetic is still within neurological grade 2 despite a significant improvement). Information about the evaluators was not specifically described in the manuscript as we were mostly interested in pragmatic outcomes looking at whether owners were happy with the patient’s improvement and quality of life (Jeffery ND, Olby NJ, Moore SA; Canine Spinal Cord Injury Consortium (CANSORT-SCI). Clinical Trial Design-A Review-With Emphasis on Acute Intervertebral Disc Herniation. Front Vet Sci. 2020 Sep 2;7:583. doi: 10.3389/fvets.2020.00583.).

- Lines 128-129: Explicitly stating the number of days is necessary, and in case of deterioration, the possibility of a patient transitioning to a different category should be addressed. For instance, if a grade 2 patient deteriorates to grade 3, the article should clarify the duration of conservative therapy before revisiting decision-making.

  • Unfortunately, as this is a retrospective study, it is not possible to provide specific timelines used to define outcome as they were not standardised at the time the cases were seen and were at the discretion of the clinician, and this is one of the limitations of such a retrospective study as of course the reviewer will be aware. We can see that the mean time between initial consultation and diagnostic imaging was 6.84 days (1 – 28) for IVDE group 2 (lines 206-207). Therefore, the sentence “At least two weeks (or less, in case of deterioration) of medical management were performed before making this decision for dogs with neurological grade 1 or 2 (ambulatory paraparetic). The timescale for reviewing decision-making was generally 5-7 days (less in case of deterioration) for dogs with neurological grade 3 and 4 (non-ambulatory dogs).” (previously lines 128-132) has now been removed as it was not used to design and carry out the study. The timeline for improvement or not was not one of the aims of the study so we do not feel this is a significant limitation, however lines 311-326 have been added to the discussion to better clarify this.

Additionally, the authors should clarify whether an MRI or CT scan was conducted in such cases.

  • All patients underwent MRI scans. This has been clarified on line 214.

- Line 132: The rationale behind waiting four or five days before performing an MRI or CT scan for a grade 4 patient needs clarification. The same query applies to grade 5 patients. Providing insights into the decision-making process and timing of scans is crucial for a comprehensive understanding.

  • Please see comment above about lines 128-129 and newly added lines 311-326. The clinical reasoning management approach described here applies to grade 1-4 dogs, as MRI and decompressive surgery are normally carried out as soon as reasonably possible on grade 5 dogs in our hospital. The fact that Grade 5 dogs were excluded from the study has now been clarified (please see lines 113-116).

In conclusion, while the article offers valuable insights, addressing the above concerns would contribute to a more comprehensive and transparent presentation of the study's findings.

  • Thank you again for your time and advice. We hope that the changes made to the manuscript and the explanations above are satisfactory, and improve the quality of the manuscript to a sufficient level for publication.

Reviewer 2 Report

Comments and Suggestions for Authors

Author Response

Thank you very much for taking the time to read and review our work and provide us with comprehensive comments aimed at improving it. Please find each of the points answered in red below.

Introduction

  • Line 55: ref 4 was not updated
    • This has been amended

Materials and methods:

  • Lack of analysis description
    • Analysis of the data that produced the results is outlined in detail within section 2 Materials and Methods. Following discussion with our statistician, we feel that further statistical analysis of the results would not add to the robustness or clarity of the study. We present descriptive statistics which are able to answer the questions the study is posing since the study design and aims were intended to be pragmatic.
  • Line 125: Improved motor function counts as successful outcome, even if remains non-ambulatory?
    • Thank you for this suggestion. All our non-ambulatory dogs improved to ambulatory status. This has been amended (lines 136-138).

  • Criteria to make the most likely diagnosis? By whom? Based on what?
    • Further information to clarify this has been added to the manuscript, please see lines 107-109

Results:

  • Unnecessary long
    • This section has been shortened by amending the following:
      • The lines about breeds have been removed (previously lines 160-162 – already described in table 2)
      • The sentence about mean age has been removed (previously line 157 – already in table 4 and described individually for dogs belonging to groups 3)
      • The sentence ‘All patients had improved voluntary pelvic limb motor function and no pain at the time of last available follow-up’ (previously line 193) has been removed as this is the definition of group 1.
      • The sentence ‘All patients had improved voluntary pelvic limb motor function and no pain at the time of the last available follow-up.’ (previously line 211) has been removed as this is the definition of group 2.
      • The sentence ‘Neurological grade of dogs that were hospitalised (mean 3, range 2 - 4) was worse than those who were not (mean 2.25, range 2 - 4).’ (previously line 239) has been removed as it is already described in table 4 or individually for dog 5
      • Follow-up time for IVDE and IM/ANNPE groups 1 has been removed from (previously) lines 192 and 248 and integrated in table 4. Follow-up time for group 2 was also added to the table.
    • How was the data regarding the actual time to define the case as successful or unsuccessful?
      • As this is a retrospective study, we can only see that the mean time between initial consultation and diagnostic imaging was 6.84 days for group 2 (range: 1 – 28) and that within IVDE group 3, both dog 1 and dog 2 were considered unsuccessful and had imaging four weeks after initial presentation. Decision-making when the case was seen was at the discretion of the clinician, hence not standardised and so impossible to report with specific details. A further description of the general rules we use for decision-making are detailed in the newly added lines in discussion 310-326.
    • Table 3: seemed not relevant to the study aim
      • This table was created as a way to summarise the history in keywords or short bullet points. We believe it is relevant in this study as history is a crucial factor in clinical reasoning and it provides information that is as important as other elements such as onset, progression etc., so we would prefer to keep it in if the reviewer agrees.
    • Figure 1 and 2: information was largely overlapped; both were not very informative
      • We think Figure 1 is a very useful visual summary of the answer to the first (‘what proportion of dogs does not improve satisfactorily and subsequently requires three-dimensional imaging?’ – group 2 and 3) and second (‘for how many of these is the final diagnosis different from the initially suspected diagnosis?’ – group 3) questions the study aims to answer. Figure 2 is a visual summary of the answer to the third question of the study (‘what is the outcome for all these patients?’) and provides an idea of the safety margin of using clinical reasoning in a population like ours. Although the figures may look similar, the information they display answers different questions and is an easy condensation of the results of the entire paper, so again we would prefer to keep both. However, if the reviewer feels strongly that both figures are unnecessary, we will of course remove them.
    • pre-referral radiographs: why include the data? Also cannot find information mentioned in the discussion (Line 269)
      • Pre-referral radiographs were included to contextualise the use of clinical reasoning in referral practice, where it is common to treat patients that have already had radiographs at the referring practice.
      • Apologies for the mistake at (previously) line 269. This has now been amended.
    • Line 191: these two cases seemed not to be suitable cases for this study aim, unless more information or radiographic information to support the DDx (IVDE may not be the most likely DDx in these patients)
      • We would like to offer further details on the cases to explain the reasons why they were included:
        • Case 1: Pug who had spinal stapling to treat a subarachnoid diverticulum; the implants had to be removed 18 months later due to implant failure. This dog was included in the study at 4 years of age as he suffered a one-week history of acute worsening of his pelvic limb gait associated with thoracolumbar pain 14 months after implants removal. This episode was considered to be unlikely to be related to the previous surgery due to the significant time period following implant removal during which the dog’s condition remained stable. The subsequent acute deterioration of gait and onset of pain led to IVDE being considered the most likely differential diagnosis.
        • Case 2: 13yo Bassett Hound cross who had a L5 spinal fracture stabilised surgically four years prior to the suspected IVDE episode. The dog presented with a 2-week history of acute paraparesis and thoracolumbar pain localising to the T3-L3 spinal cord segments following almost 4 years of stable gait and no back pain, and hence again thoracolumbar IVDE was considered the most likely differential diagnosis.
      • A short description of these cases (lines 290-298) was also added to the manuscript to be transparent with the reader. The authors are happy to create a table with more information about all these cases if deemed appropriate by the reviewer.

Discussion

  • Line 257: Biased statement (the majority of cases were with low neurological grade)
    • This has been changed to ‘in many ambulatory dogs’.
  • Line 269: pre-rads = inconclusive? I could not find relevant data in the results section.
    • The word ‘inconclusive’ has been removed from line 270 (previously 269).
  • Line 284: “feasible” seemed not appropriate
    • This has been changed to ‘possible’.
  • Line 288: same comments as for Line 191.
    • Please see comment above.
  • Line 301: how about if remained static in the following day?
    • Please see the newly added lines 311-326.

We hope that these changes have improved the quality of the manuscript and clarified the aims and methodology as well as reporting of results.

Reviewer 3 Report

Comments and Suggestions for Authors

The manuscript "Outcome of the use of clinical reasoning alone in dogs with acute thoracolumbar myelopathy and/or pain" presents scientific soundness and relevance. The article is well written and it bring new information to the Animals's readers. 

In general, I found few situations needing correction. Please, find it below:

a) Introduction and Conclusion. Lines 82-87 - The second objetive of the study (For how many of these is the final diagnosis different from the initially suspected diagnosis?) need to be better answered in the section "Conclusion" (lines 334-345).

b) Introduction. I suggest that the authors add a few sentences to the first paragraph  explaining the frequency, relevance, epidemiology and clinical signs of conditions related to thoracolombar intervertebral disc disease in dogs.

Author Response

The manuscript "Outcome of the use of clinical reasoning alone in dogs with acute thoracolumbar myelopathy and/or pain" presents scientific soundness and relevance. The article is well written and it bring new information to the Animals's readers. 

In general, I found few situations needing correction. Please, find it below:

a) Introduction and Conclusion. Lines 82-87 - The second objetive of the study (For how many of these is the final diagnosis different from the initially suspected diagnosis?) need to be better answered in the section "Conclusion" (lines 334-345).

  • Thank you very much for taking the time to review our work and for your kind and valuable comments. The sentence ‘When patients underwent MRI due to deterioration, the final diagnoses were different from the initially suspected IVDE diagnosis in only 2/21 cases.‘’ (lines 365-366) was added to the conclusions to provide more information about the second question of the study.

b) Introduction. I suggest that the authors add a few sentences to the first paragraph  explaining the frequency, relevance, epidemiology and clinical signs of conditions related to thoracolombar intervertebral disc disease in dogs.

  • More information about IVDE, IM and ANNPE was added at lines 47-51 and 69-72. We hope that these corrections are comprehensive enough to have improved the manuscript and warrant publication.

Round 2

Reviewer 1 Report

Comments and Suggestions for Authors

No more comments

Author Response

Thank you for your time and for accepting the manuscript for publication

Reviewer 2 Report

Comments and Suggestions for Authors

One dog had 2 events. Throughout the text, sometimes the denominator counted this patient as 1, and other times as 2. Sometimes, the description was confusing, or the analysis did not make sense. For example:

Line 162: should this be 142 events?

Line 167: were there 139 or 140 dogs? The gender description seemed to count that dog twice, but in Table 2 (breeds), it seemed to count it only once.

Line 179: 123 dogs? So, was the dog with two events counted for only one event?

Please check the manuscript thoroughly and correct the data or state it accurately. If I interpreted incorrectly, please improve the writing to make it clear.

Table 3: for the IVDE group, some owners mentioned multiple complaints for the same patient, so use 226 as the denominator. However, this analysis was less informative than using 123 dogs (or 124 events?) as the denominator. Particularly, the statement in Line 267 about 4.8% (11/226) was incorrect or misleading. Table 3 appeared more suitable for supplemental data.

Figures 1 and 2:

Both figures duplicated the text, not complemented it. Furthermore, based on your definition of the outcome groups, the information overlapped completely between these two figures. If authors strongly felt they were useful summaries, perhaps consider combining them into one figure.

Line 323: ref 44 seemed not relevant to your study and statements (did not include dogs without deep pain sensation in the present study)

Line 359-363: it should be emphasized again that the conclusion was based on a population where most dogs presented with mild clinical signs.

Line 179: 81-95%... The data expression style was difficult to read (but I am unsure whether the journal requests this).

Author Response

One dog had 2 events. Throughout the text, sometimes the denominator counted this patient as 1, and other times as 2. Sometimes, the description was confusing, or the analysis did not make sense. For example:

Line 162: should this be 142 events?

Line 167: were there 139 or 140 dogs? The gender description seemed to count that dog twice, but in Table 2 (breeds), it seemed to count it only once.

Line 179: 123 dogs? So, was the dog with two events counted for only one event?

Please check the manuscript thoroughly and correct the data or state it accurately. If I interpreted incorrectly, please improve the writing to make it clear.

  • Apologies for this oversight and thank you for pointing it out. We have now amended the incorrect numbers and modified lines 162-169 to provide clarity on the dog who had two occurrences and the total number of dogs included in the study. Table 2 contains 139 cases.

Table 3: for the IVDE group, some owners mentioned multiple complaints for the same patient, so use 226 as the denominator. However, this analysis was less informative than using 123 dogs (or 124 events?) as the denominator. Particularly, the statement in Line 267 about 4.8% (11/226) was incorrect or misleading. Table 3 appeared more suitable for supplemental data.

  • Thank you very much for this suggestion. We agree that using 123 dogs as the denominator is indeed more informative and more truly reflective of the data we wanted to report, so this has been changed on the table and line 267 has also been amended.

Figures 1 and 2:

Both figures duplicated the text, not complemented it. Furthermore, based on your definition of the outcome groups, the information overlapped completely between these two figures. If authors strongly felt they were useful summaries, perhaps consider combining them into one figure.      

  • Figure 1 and 2 have been combined into a new Figure 1, as suggested. Figure 3 (now figure 2) has also been slightly modified to match the styles.

Line 323: ref 44 seemed not relevant to your study and statements (did not include dogs without deep pain sensation in the present study)

  • This reference has now been removed.

Line 359-363: it should be emphasized again that the conclusion was based on a population where most dogs presented with mild clinical signs.

  • This has been modified to: ‘Ambulatory dogs with a clinical picture which generates IVDE as the most likely differential diagnosis have around an 80% chance of having a successful outcome when treated medically for suspected IVDE without the aid of cross-sectional imaging’.

Line 179: 81-95%... The data expression style was difficult to read (but I am unsure whether the journal requests this).

  • This has been modified to: ‘Of these patients, 100/123 (81%, with 95% confidence interval (CI) 73 - 87) belonged to […]’.